# Evaluation of Macular and Optic Disc Radial Peripapillary Vessel Density Using Optical Coherence Tomography Angiography in Gout Patients

**DOI:** 10.3390/diagnostics13243651

**Published:** 2023-12-12

**Authors:** Özgür Eroğul, Adem Ertürk, Mustafa Doğan, Kudret Kurt, Murat Kaşıkcı

**Affiliations:** 1Department of Ophthalmology, Faculty of Medicine, Afyonkarahisar Health Sciences University, 03030 Afyonkarahisar, Turkey; mustafadogan@yahoo.com (M.D.); kudret0595@gmail.com (K.K.); 2Division of Rheumatology, Department of Internal Medicine, Faculty of Medicine, Afyonkarahisar Health Sciences University, 03030 Afyonkarahisar, Turkey; adem.erturk@afsu.edu.tr; 3Department of Ophthalmology, Mugla Training and Research Hospital, 48000 Mugla, Turkey; drmuratk10@gmail.com

**Keywords:** gout, uric acid, optical coherence tomography angiography, capillary plexus density, choriocapillary flow deficit

## Abstract

In this cross-sectional study, optical coherence tomography angiography (OCT-A) findings were compared in patients with gout (*n* = 30) and healthy participants (*n* = 32). The superficial and deep vessel density variables measured using OCT-A were compared between the groups. The superficial foveal and perifoveal vessel densities of the patient group were lower than those of the healthy participants (*p* = 0.014 and *p* = 0.045, respectively). However, all superficial and parafoveal vessel densities were similar in both groups (*p* = 0.469 and *p* = 0.284, respectively). The deep capillary plexus density measurements of the whole-zone, foveal, parafoveal, and perifoveal vessel densities using OCT-A revealed no significant differences between the groups (*p* = 0.251, *p* = 0.074, *p* = 0.177, and *p* = 0.881, respectively). A higher serum uric acid (SUA) level was found to be independently associated with a decreased superficial capillary plexus density and an increased choriocapillary flow deficit in the study population. Men were less sensitive to high SUA levels than women. These findings suggest that an elevated uric acid concentration may play a role in the development and progression of cardiovascular disease through changes in the microvasculature, as shown by the OCT-A parameters.

## 1. Introduction

Gout, also known as podagra disease, is a clinical condition characterized by recurrent acute inflammatory arthritis and manifests as joint swelling that is red, tender, and accompanied by a sensation of heat. When gout affects the big toe, it is called podagra disease [1]. The most frequently affected zone is the instep-finger bone joint at the base of the big toe, which accounts for 50% of cases. However, gout may also present as calcification, kidney stones, or urate nephropathy. Gout is caused by increased blood uric acid levels. The uric acid crystallizes and accumulates in the joints, tendons, and surrounding tissues.

Gout is clinically diagnosed when uric acid crystals are identified in the joint fluid. Treatment with nonsteroidal anti-inflammatory drugs, steroids, or colchicine can reduce the symptoms. After an acute attack resolves, the uric acid levels usually decrease with lifestyle changes. Allopurinol or probenecid provide long-term protection in individuals with frequent attacks despite treatment. In recent years, the incidence of gout has increased, affecting approximately 1–2% of the western population. This increased incidence can be attributed to the increase in risk factors such as a longer life expectancy, metabolic syndrome, and changes in diet in the population.

Gout may appear in several forms, but the most common form is recurrent attacks of acute inflammatory arthritis (hot, tender, red, and swollen joints) [2]. The metatarsal-finger bone joint at the base of the big toe is the most affected joint, accounting for half of cases. Other joints such as the heel, knee, wrist, and finger may also be affected [3]. Joint pain usually begins at night and lasts for approximately 2–4 h [3]. The reason for this is a low body temperature [1]. Other symptoms such as fatigue and high fever may occur, but rarely with joint pain [1,3]. Long-standing high uric acid levels (hyperuricemia) can also cause other symptoms, such as calcification, i.e., the hard, painless accumulation of uric acid crystals. Extensive calcification may cause chronic arthritis due to bone erosion [4]. Furthermore, high uric acid levels lead to the precipitation of crystals in the kidneys, leading to stone formation and, subsequently, to urate nephropathy [5].

The underlying cause of gout is hyperuricemia. Hyperuricemia may occur due to several reasons, including genetic predisposition, diet, or the insufficient excretion of urate, the uric acid salts [6]. Approximately 10% of people with hyperuricemia develop gout at some point in their lives [7]. However, the risk varies depending on the level of hyperuricemia, with a risk of 0.5% per year for individuals with levels between 415 and 530 µmol/L (7 and 8.9 mg/dL) and 4.5% per year for individuals with levels higher than 535 µmol/L (9 mg/dL) [7].

Gout is a crystal deposition disease that occurs due to the oversaturation of body tissues with urate and the accumulation of monosodium urate (MSU) crystals in and around joints. Gout is the most common inflammatory arthritis disease in men and leads to a serious deterioration in the patient’s quality of life [8]. Clinical manifestations include acute gouty arthritis attacks with excruciating pain, chronic joint damage due to MSU deposition in the joints, kidney stone formation, and kidney failure.

Optical coherence tomography angiography (OCT-A) is a novel diagnostic tool used as an alternative or adjunct to classical fluorescein angiography (FA). OCT-A, a rapid and noninvasive imaging tool, acquires an image of the retinal vasculature by sensing blood flow. In contrast to classical FA, it is a non-staining procedure and does not lead to any fluorescein-related side effects such as vomiting, hypersensitivity reactions and cardiovascular complications [9]. This technology enables the high-resolution in situ visualization of individual vascular layers. In addition, OCT-A helps to visualize the deep, superficial and choroidal vascular network and can even identify the mid-capillary plexus; this is unlike FA, which only shows the superficial capillary network [10].

This study aimed to evaluate microangiopathic changes using OCT-A in patients with gout.

## 2. Materials and Methods

In this cross-sectional study, data were collected after OCT-A measurements were performed in 30 eyes (right eyes) of 30 patients with gout and 32 eyes (right eyes) of 32 healthy individuals with no systemic and eye disease. This study was conducted between January 2021 and May 2022 in the Division of Rheumatology and Department of Ophthalmology. All patients underwent a clinical examination and ophthalmological examination at the first and last visit. Individuals who had visual acuity measurements and corneal and fundus examinations were included in the study. Patients with gout were included in the study as the patient group, and those without gout or any other systemic disease and who were completely healthy in terms of ophthalmology were included in the study as the control group. Those with a vascular pathology associated with a systemic disease other than gout and glaucoma, cataract, retinopathy, keratopathy, congenital ocular anomaly and who had undergone ocular surgery within 6 months, those with systemic vascular disease including diabetes and systemic hypertension, those with a long history of treatment for hypertension, pregnant or lactating women, and those with pupil dilation or hypersensitivity/intolerance to topical anesthetics or mydriatics were excluded from the study.

All participants underwent a complete ophthalmic examination, including slit lamp examination, best-corrected visual acuity, a measurement of intraocular pressure (IOP) with Goldman Applanation Tonometry, and a fundus examination. B-scan ultrasonography was performed to evaluate the orbital and ocular structure.

The study was conducted in accordance with the principles of the Declaration of Helsinki, the method and purpose of the study were explained in detail to all participants, and informed consent was obtained from each participant.

### 2.1. Optical Coherence Tomography Angiography (OCT-A) Measurements

OCT provides a detailed non-invasive view of the structure of the retina. OCT-A enables the visualization of 3D retinal and choroidal microcirculatory maps without the use of exogenous stains by simultaneously visualizing the inner and outer retinal blood flow [11,12]. In particular, the macular vessel density and the size of the foveal avascular zone (FAZ) are known to be very important for visual acuity. More recently, OCT-A has been used for the diagnosis and monitoring of retinal microvascular pathology by measuring the macular vessel density in patients with diabetic retinopathy, central serous chorioretinopathy and glaucoma, as well as for the assessment of macular oedema, age-related macular degeneration and other conditions [12,13,14].

OCT-A measuremenst were obtained from all participants after pupil dilation. Drops comprising 25 mg/mL of phenylephrine hydrochloride (Mydfirin, Alcon, Fort Worth, TX, USA) and 1% tropicamide (Tropamid^®^ 1% Forte, Bilim Pharmaceuticals, Istanbul, Turkey) were used for dilation. OCT-A images were acquired from a 4.5 × 4.5 mm optic nerve-centered area using Optovue AngioVue™ (RTVue XR Avanti, Optovue Inc., Fremont, CA, USA) and AngioAnalytics using the Split-Spectrum Amplitude Decorrelation Angiography algorithm. Analysis was performed using the 2.0 quantification software. The wavelength was 840 nm, the scan frequency was 70,000 Hz, the lateral and axial resolutions were 15 µm and 5 µm, respectively, the scan depth was 2–3 mm, the number of A-scans was 304 × 304, and B-scans were repeated twice in the same spot. The Motion Correction Technique and DualTrac were used throughout, and the HD Angio Disc 4.5 mm mode was used to scan a 4.5 × 4.5 mm zone around the optic nerve.

Images of four layers were obtained for each patient: superficial capillary plexiform (SCP), deep capillary plexiform (DCP), outer retina (OR), and choriocapillaris (CC). Macular OCT-A images were acquired with AngioPlex (Carl Zeiss Meditec, Dublin, CA, USA) using a Cirrus high-resolution OCT prototype. A 6 × 6 mm scan model was used to image the macula.

### 2.2. Statistical Analysis

The data were analyzed using the IBM SPSS Statistics 23 (SPSS Inc., Chicago, IL, USA) package program. The data were summarized and described using numbers, mean values, standard deviations, the median, and interquartile range (IQR). The Kolmogorov–Smirnov test was used to test whether the data followed a normal distribution. The independent sample *t*-test and Mann–Whitney U test were used to evaluate the differences between variables. The relationships between variables were assessed using the Pearson correlation test. The results were evaluated at a 95% confidence interval and 5% significance level.

## 3. Results

In the present study, 62 eyes of 62 cases, including 32 healthy participants and 30 patients with gout, were analyzed. There was no significant difference in terms of gender between the control group and the patient group. The control group consisted of 12 women and 20 men. There was a higher male population in both groups. However, the BMI was similar in the patient and control groups. The average BMI was between 21–25. Thirty participants (10 women and 20 men) who had at least one gout attack and recovered with treatment and did not have any previous systemic or ophthalmic diseases were included. No differences in age and shooting quality were detected between the two groups (*p* = 0.451 and *p* = 0.468, respectively). However, regarding gender distribution, males were higher in the patient group. The mean age of the patient group was 58 ± 17 years and that of the healthy group was 57 ± 18 years (Table 1).

No significant difference was detected in terms of the spherical equivalent and axial length in the patient and control groups between the cases participating in the study. The average spherical equivalent of the control group was +0.50 ± 0.75 and the axial length was 23.50 ± 1.25 mm. On the other hand, the average spherical equivalent of the patient group was +0.75 ± 100 and the axial length was 23.25 ± 1.75 (*p* = 0.242 for spherical equivalent, *p* = 0.228 for axial length). The IOP values in the patient and control groups were 12.5 ± 4.8 mm/hg and 13.1 ± 4.4 mm/hg, respectively.

Table 2 shows the OCT-A findings for the patients with gout and healthy participants. The vessel densities of both the superficial and deep capillary plexuses were lower in the patients with gout than in the healthy participants. The superficial foveal and perifoveal vessel densities of the patient group were lower than those of the healthy participants (*p* = 0.014 and *p* = 0.045, respectively). However, all superficial and parafoveal vessel densities were similar in both groups (*p* = 0.469 and *p* = 0.284, respectively). The DCP measurements of the whole-zone, foveal, parafoveal, and perifoveal vessel densities using OCT-A revealed no significant differences between the groups (*p* = 0.251, *p* = 0.074, *p* = 0.177, and *p* = 0.881, respectively) (Table 2, Figure 1a–d,a*–d*).

Table 3 shows the density measurements of the FAZ and perimeter fovea at the 300-degree circumferential zone. Significant differences were detected between the patient and control groups in the measurements of the FAZ, perimeter, and 300-degree peripheral zone of the fovea (*p* = 0.000, *p* = 0.011, and *p* = 0.024, respectively).

Table 4 compares the patient and control groups in terms of the flow zone (FA) parameters in the choriocapillaris and OR layer. Significant differences were found between the patient and control groups in terms of the choriocapillaris flow zone (*p* = 0.008), as the choriocapillaris flow zone value was found to be significantly lower in the patient group compared to the healthy group. On the other hand, no significant differences were found between the groups in terms of the OR flow zone (*p* = 0.464). (Table 4, Figure 2a,b,a*,b*).

The OCT analysis and OCT-A results are shown in Table 5. No significant differences in the mean RNFL were found between the groups (*p* = 0.693). However, the C/D ratios were different between the groups (*p* = 0.000). In terms of C/D values, the values of the patient group were higher than those of the healthy controls. Based on the ICT-A measurements, significant differences in vessel densities were found between the groups, as the whole vessel density and peripapillary vessel density were significantly decreased in the patient group compared to the control group (*p* = 0.014 and *p* = 0.045, respectively). On the other hand, only inside vessel densities were found to be similar between the groups (*p* = 0.278).

Correlation analysis in the patient group revealed a strong, positive, and moderately linear relationship among all variables.

Table 6 shows the SCP, DCP, and CFD values according to the SUA levels in male and female participants. Male patients are less sensitive to SUA levels than female participants.

## 4. Discussion

Due to its non-invasiveness, capacity for rapid scanning and high resolution, OCT-A is widely used in clinical trials and practice. This study aimed to evaluate the effects of serum uric acid levels on chorioretinal microcirculation in patients with gout, independent of other traditional cardiovascular risk factors. In the presence of high serum uric acid levels (SUA), low foveal and perifoveal SCP and low peripapillary VD were detected, which were more significant in females. However, although the whole-zone and parafoveal SCP, all DCP densities, and RNFL values were lower in the patient group than in the control group, this difference was not statistically significant. In addition, the SUA level was positively correlated with CFD. To the best of our knowledge, this is the first study to investigate the effects of gout on the chorioretinal microvasculature on the basis of OCT-A.

Gout can also affect the retinal and choroid vessels. Due to the long-term complications of hyperuricemia and its close association with vascular diseases [15], this study also aimed to investigate the macular vessel density and FAZ measurements.

Previous studies that focused on hyperuricemia-related changes in retinal ganglion cells and RNFL did not reveal a significant difference in gout patients when compared to healthy controls, and the RNFL was not different in gout patients [16,17], consistent with our study.

In the present study, low SCP values were observed in response to high SUA levels. A reduced SCP may indicate early systemic vascular injury. There is a strong association between retinal vascular disease and cardiometabolic events. Some epidemiological studies have shown that changes in the SUA over time are a significant risk factor for type 2 diabetes, high blood pressure and a rapid decline in kidney function [18,19]. In the study of healthy adults by You et al., SCP was associated with male gender, older age and a longer AL. In addition, they reported that a lower DCP was significantly associated with a longer AL and higher creatinine levels [20]. This present study enhanced prior research by adding the SUA level to the analysis. On the basis of our findings, the correlation between SUA and cardiovascular risk may be explained in part by the change in the microvascular system caused by SUA.

CC plays an important role in nourishing the OR and supplying oxygen. Therefore, CC changes are effective in the etiopathogenesis of diabetic retinopathy, age-related macular degeneration, pathological myopia, and central serous chorioretinopathy. CFD was also found to be a predictor of outcomes in patients with posterior uveitis, and CC abnormalities were observed in systemic vascular disease and pregnancy complications. As a result, CC levels can provide significant information on the pathogenesis of ocular and systemic diseases. A study by Cheng et al. showed that blood pressure, lipid levels and age influence CFD in OCT-A. This study, however, did not analyze SUA. After adjusting for other confounding factors in this study, high SUA levels in both men and women were found to be associated with an increase in the CFD. Thus, SUA should be considered in the interpretation of CC changes in normal and diseased individuals.

High SUA levels led to a decrease in vessel density in the whole zone and peripapillary zone. However, a significant increase in the C/D ratio was found in patients with high SUA levels. This result is consistent with other studies in the literature model [21].

In the present study, FAZ, the perimacular zone, and the 300 μm wide FAZ FD-300 values that represent the VD of the whole retina were found to be significantly reduced compared to healthy volunteers. Thus, these values were lower in patients with gout.

High SUA levels may be responsible for gout’s association with an impaired retinal capillary plexus and CC. First, high SUA levels can cause hypertension. In the study conducted by Kuwata et al., high SUA prevalence rates of 25%, 60%, and 90% were reported in adolescents with new-onset hypertension and in patients with untreated primary hypertension [22]. In addition, chronic hyperuricemia, the activation of the renin–angiotensin system, and the inhibition of nitric oxide synthetase may cause the narrowing of the renal vasculature, thereby increasing the risk of atherosclerosis and hypertension. Both high SUA levels and the use of Allopurinol in the treatment of gout can induce endothelial dysfunction. Endothelial dysfunction was found to be caused by impaired nitric oxide synthesis in a hyperuricemic animal model [23]. The SUA level can stimulate oxidative stress, cytokine secretion, and inflammation. Following treatment to reduce SUA levels, there is an increase in the production of reactive oxygen species by the endothelial cells due to the activation of NADPH oxidase [24]. In a previous study, the SUA level was associated with vascular regeneration and the proliferation of vascular smooth muscle cells, which are involved in the etiology of atherosclerosis [25].

Previous studies on gender associations of SUA with diabetes mellitus and its complications have been documented [22,26,27]. Although both genders have an increased risk of developing diabetes mellitus with hyperuricemia, women with hyperuricemia were found to be more likely to develop diabetes than men with hyperuricemia [28]. In a previous study, it was shown that an increased risk of new-onset diabetic retinopathy in male patients was associated with a high level of SUA. However, it was stated that this was not the case in female patients [22]. High SUA levels may be more harmful in women than in men, according to other studies [27,29,30,31]. A Chinese study reported that each standard deviation increase in SUA was associated with a 1.68 µm increase in retinal venous caliber in women, but not in men [32]. In this study, microvascular vessel densities in terms of high SUA levels were found to be more significant in females, as in previous studies. It can be concluded that women are more sensitive to high levels of SUA than men. However, to clarify the effect of gender on SUA and OCT-A parameters, further studies with more participants are needed.

A standard protocol, homogeneous population, and the use of a state-of-the-art OCT-A device are the strengths of our study. The study is therefore an important addition to the current quantitative data on the normal retinal vasculature. However, it is also important to note the limitations of the study. Firstly, the SUA data were obtained using only one test, but there might be variations in the SUA concentration throughout the year. However, it is not pragmatically possible to test SUA levels in a population more than once. It would have been more informative if baseline and long-term changes in the SUA levels and OCT-A parameters were available. Moreover, information on the details of the drug use of the participants was not collected. The use of these drugs might also affect retinal VD measurements [33,34]. In addition, it was not possible to evaluate the daily variations in the vessel density between hyperuricemic and normal participants. This study was conducted in a population of Turkish origin. Further studies are therefore necessary for confirmation of the results in other populations.

## 5. Conclusions

In the present study, a higher SUA level was independently associated with an increased CFD and decreased SCP. Men were less sensitive to high SUA levels than women. These findings suggest that elevated UA levels play a role in the development and progression of cardiovascular disease via microvascular changes, as demonstrated by the OCT-A parameters. Further studies are needed to clarify the underlying pathophysiology and determine whether SUA-targeted treatment improves chorioretinal microcirculation.

## Figures and Tables

**Figure 1 diagnostics-13-03651-f001:**
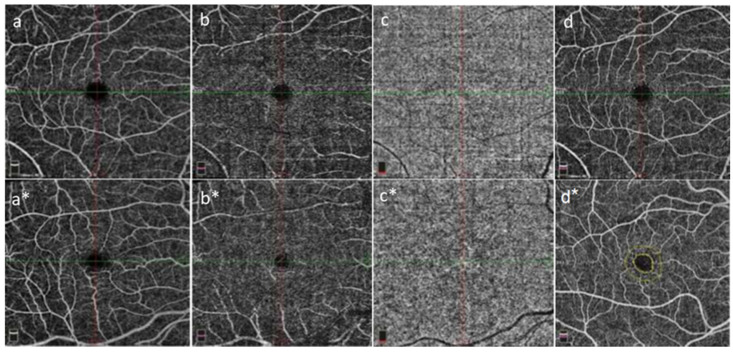
GoutT patients (images with asterisks) and healthy participants (images without asterisks), respectively; superficial capillary plexus (**a**,**a***), deep capillary plexus (**b**,**b***), choriocapillaris (**c**,**c***) and retina (**d**,**d***); OCT-A images are available. It is observed that gout patients (**a***,**b***) have lower vascular densities in the superficial and deep capillary plexuses compared to healthy participants (**a**,**b**).

**Figure 2 diagnostics-13-03651-f002:**
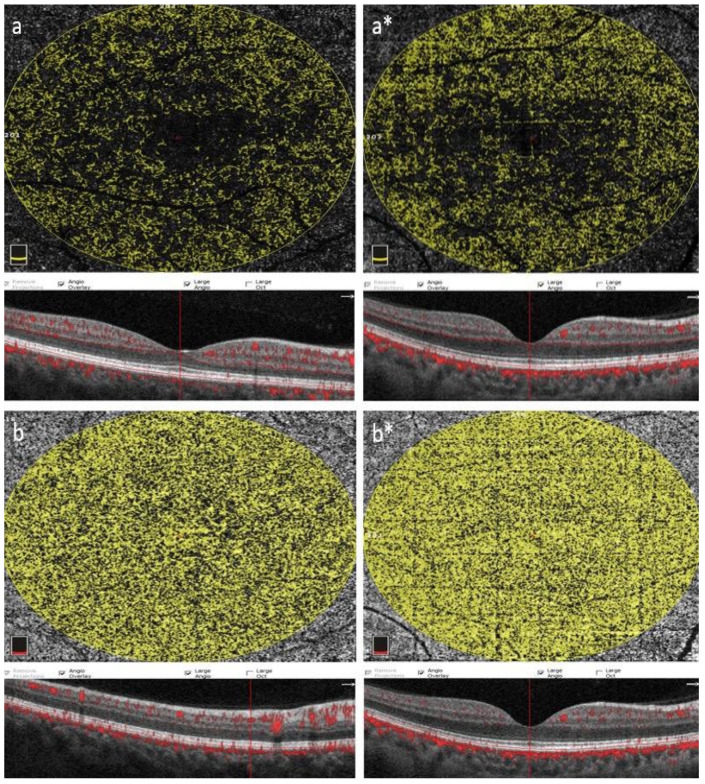
En face optical coherence tomography angiograms of the outer retina and choriocapillary flow field (6.0 × 6.0 mm scan size and 21,056 mm^2^ flow field, scan quality index = 8/10) are shown for patients with gout and healthy controls. Outer retina (**a**,**a***) and choriocapillaris (**b**,**b***) segments of the right eye of patients with gout and healthy controls (GUT patients = letters without an asterisk and healthy controls = letters with an asterisk) and the foveal center of patients with gout and healthy controls. Corresponding OCT slices throughout are shown below each figure section. The patient and control groups were similar in terms of the outer retinal flow area (*p* = 0.464), but a significant impairment was detected in the patient group in terms of the choriocapillaris flow area (*p* = 0.008).

**Table 1 diagnostics-13-03651-t001:** Demographic features of study groups.

Variables	Gout Patients (30)	Control Group (32)
Age	58 ± 17	57 ± 18
Female/Male	10/20	12/20
City/Village	22/8	17/15
Working/Non-working	24/6	19/13
Married/Single	26/4	29/3

**Table 2 diagnostics-13-03651-t002:** Comparison of control and patient groups in terms of retinal superficial and deep vessel density variables measured using OCT-A.

Variables-OCT-A (VD) (mm^−1^)	Patient (30)	Control (32)	*p*-Value
SCP WHOLE	51.61 ± 3.69	52.28 ± 3.58	0.469
SCP FV	19.92 ± 6.71	23.80 ± 5.26	0.014
SCP PARAFV	54.30 ± 3.6	54.75 ± 4.3	0.284
SCP PERIFV	54.60 ± 3.2	52.95 ± 4.30	0.045
DCP WHOLE	55.80 ± 3.80	56.700 ± 7.10	0.251
DCP FV	37.29 ± 7.60	40.79 ± 7.51	0.074
DCP PARAFV	58.00 ± 5.00	59.60 ± 5.70	0.177
DC	58.70 ± 3.50	59.05 ± 7.10	0.881

**Table 3 diagnostics-13-03651-t003:** Comparison of patient and control groups in terms of FAZ variables.

Variables—FAZ Parameters (mm^2^)	Patient (30)	Control (32)	*p*-Value
FAZ	0.21 ± 0.05	0.29 ± 0.09	0.000
Perimeter	1.90 ± 0.26	2.13 ± 0.40	0.011
FD-300 (%)	55.17 ± 3.44	57.19 ± 4.20	0.024

**Table 4 diagnostics-13-03651-t004:** Comparison of control and patient groups in terms of the flow zone–FA variables.

Variables (FA)	Patient (30)	Control (32)	*p*-Value
OR FA	8.50 ± 1.81	8.41 ± 2.51	0.464
CC FA	19.07 ± 1.26	19.88 ± 1.08	0.008

**Table 5 diagnostics-13-03651-t005:** Comparison of control and patient groups in terms of optic disc variables.

Variables (Optic Disk)	Patient (30)	Control (32)	*p*-Value
RNFL GLOBAL	113.60 ± 11.09	114.72 ± 11.08	0.693
WHOLE VD	48.28 ± 2.36	49.73 ± 2.15	0.014
INSIDE VD	50.89 ± 4.58	52.26 ± 5.26	0.278
PERIPAPILLARY VD	50.52 ± 3.73	52.16 ± 2.51	0.045
CUP-DISC RATIO	0.17 ± 0.16	0.05 ± 0.03	0.000

**Table 6 diagnostics-13-03651-t006:** Analysis of the relationships between SUA levels (Every 10 μmol/L Increase) and SCP, DCP, and choriocapillary flow deficit (CFD) in men and women separately.

Variables	Female	*p*-Value	Male	*p*-Value
SCP	−0.152	<0.001	0.003	0.857
DCP	−0.047	0.02	0.008	0.612
CFD	0.025	<0.001	0.01	0.145

## Data Availability

Detailed data are available upon request from the corresponding author.

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
