# Peer review of "Evaluation of Macular and Optic Disc Radial Peripapillary Vessel Density Using Optical Coherence Tomography Angiography in Gout Patients"

_diagnostics, 2023, doi:10.3390/diagnostics13243651_

Round 1

Reviewer 1 Report

Comments and Suggestions for Authors

This study used OCTA to evaluate vascular density in patients with gout. The manuscript is interesting, but several comments need to be addressed before publication can be considered. In this study, ophthalmologic examination is insufficient.

1.       As described in the "Discussion", SUA may be related to gender. Therefore, it should be shown that there is no significant difference in gender between the gout patients group and the healthy participants group. Also, the results need to be indicated for BMI.

2.       Refractive error and axis length are involved in retinal and choroidal morphology and circulatory dynamics. Therefore, data on both should be presented and differences between the two groups should be investigated.

3.       In order to clarify the effect of gout (SUA) on the microvascular circulation of the retina, it is useful to examine the oscillatory potentials in the electroretinogram. In addition, data on retinal and choroidal thickness should also be shown to describe the reduction in retinal and choroidal vascular density.

4.       As the authors discuss in their "Discussion", there is an association between high SUA and blood pressure. Therefore, the blood pressure of gout patients and healthy participants in this study should be presented to determine if there are any differences between the two groups, and to investigate the correlation with OCTA data.

5.       Please provide the rationale for the use of both parametric and non-parametric tests in the statistical examination of this study.

Author Response

Dear Editor

We would like to thank you and the referees for your comments and suggestions that have all been taken into consideration in the revised manuscript that we are now ready to submit. Please find listed below the detailed answer to all the comments and criticisms underlined by the referees. We hope the revised manuscript is now suitable for consideration in your eminent journal(The content in the main text has been edited to 4040 words).

Reviewer 2 Report

Comments and Suggestions for Authors

I would like to congratulate the authors on their work. A few issues that need improvement are:

1. In the OCT-A measurement section instead of providing some general information about OCTA you need to clarify which OCTA equipment you used (e.g. Heidelber Spectralis, Optovue) which protocol for scanning (e.g. 20 arc 3x3mm with 97 line scans). Also clarify if patients were dilated, which drops did you use etc.

2. In the results section it would be better if you provided a table with the demographic parameters of cases and controls as table 1.

3.Did you measure the IOP of the participants as this could have also influenced the RNFL results?

4. One of the key findings of the study is the decreased vessel density in the foveal region in both superficial and deep capillary plexi, and also the reduced foveal avascular zone area in patients with gout. One would expect that an impairment in the microvascular system to lead to an enlarged foveal avascular zone as a sign of macular ischemia.

5. Did you measure any other foveal avascular zone parameters such as the perimeter or the circularity index? This could also be of scientific interest in this topic.

Author Response

(The authors gave the same response as above.)

Round 2

Reviewer 1 Report

Comments and Suggestions for Authors

Thank you for your response.

I hope this study will contribute to the future development of medicine.